# The Longitudinal Relationship between Cyberbullying Victimization and Loneliness among Chinese Middle School Students: The Mediating Effect of Perceived Social Support and the Moderating Effect of Sense of Hope

**DOI:** 10.3390/bs14040312

**Published:** 2024-04-11

**Authors:** Jing Wu, Xu Zhang, Qianxiu Xiao

**Affiliations:** 1Department of Psychology, Shaoxing University, Shaoxing 312000, China; 21020451089@usx.edu.cn (X.Z.); 22020451037@usx.edu.cn (Q.X.); 2School of Educational Sciences, Ludong University, Yantai 264025, China

**Keywords:** cyberbullying victimization, loneliness, sense of hope, social support, moderated mediation model, longitudinal study

## Abstract

Compared with traditional forms of bullying (e.g., physical bullying, verbal bullying), cyberbullying victimization can bring heavy psychological damage to the victim of bullying. Studies have found that cyberbullying victimization leads to higher levels of depression and causes anger and emotional problems. Nevertheless, existing studies mainly focus on traditional bullying while affording scant consideration to the longitudinal impact of cyberbullying on mental well-being. The purpose of this study was to examine the effects of cyberbullying victimization on middle school students’ loneliness while simultaneously investigating the mediating role of perceived social support and the moderating role of feelings of hope. A total of 583 middle school students were surveyed using four self-report questionnaires. Cyberbullying victimization predicts loneliness. Perceived social support mediates the role of cyberbullying victimization in influencing cyberbullying. Sense of hope moderated the direct pathway and the second half of the mediating role pathway. First, many mediating and moderating variables of cyberbullying victimization affect loneliness, and different mediating and moderating variables can be studied in the future. Second, future studies could expand this study’s sample to validate the results of this study. Third, this study only collected data at two time points, and future studies could collect data at multiple time points. Cyberbullying victimization can increase loneliness over time. Perceived social support and a sense of hope can mitigate the effects of cyberbullying victimization on an individual’s mental health.

## 1. Introduction

As information technology continues to advance, minors’ internet usage has progressively risen, leading to a deepening dependence on online platforms. According to the Research Report on National Minors’ Internet Usage in 2021 released by China Internet Network Information Center [1], there is a noticeable trend in China toward younger ages for minors’ internet usage, with a penetration rate among minors reaching 96.9%. A survey on the Internet safety of young people showed that the proportion of youths who had suffered from cyberbullying was as high as 71.11% [2]. Cyberbullying manifests in various forms, such as online ridicule, sarcasm, verbal abuse, intimidation with insulting words and pictures, and so on. It is evident that along with the increasing popularity of the Internet, there is a concurrent rise in undesirable online behaviors, such as cyberbullying.

### 1.1. Cyberbullying Victimization and Loneliness

Compared to traditional forms of bullying, such as physical and verbal aggression, cyberbullying victimization may not inflict direct physical harm on the victim, but it can lead to severe psychological distress, triggering a cascade of negative emotions. Some studies have found that cyberbullying victimization leads to higher levels of depression in individuals compared to traditional bullying victimization [3,4]. The experience of cyberbullying victimization can damage adolescents’ self-esteem and self-perception and cause anger [5] and emotional problems [6]. Existing studies have demonstrated that cyberbullying victimization is associated with a decline in adolescents’ academic performance [7]. Moreover, it triggers a range of psychological problems such as anxiety, insomnia [8], depression [9], and even suicidal ideation and behavior [10,11]. Therefore, cyberbullying victimization may increase adolescents’ loneliness.

### 1.2. Social Support as a Mediator

Cyberbullying victimization is essentially a problem with an individual’s interpersonal relationships in virtual space [12]. It has been shown that cyberbullying victimization reduces an individual’s desire to interact with others [13] and impairs relationship satisfaction. According to self-determination theory, individuals’ basic psychological needs are categorized into autonomy, competence, and belongingness. The fulfillment of basic psychological needs promotes mental health development and increases the level of individual well-being [14]. When individuals’ basic psychological needs are unmet, they tend to experience more negative emotions [15], which may lead to depression [16] and suicidal ideation [17]. Social support is a supportive resource for individuals to maintain their mental health via access to social connections. Social support can be categorized into two types: objective social support; and perceived social support. Perceived social support refers to people’s subjective evaluation of the degree of support they receive from significant others [18]. Perceived social support not only predicts and promotes healthy development [19] but also serves as a significant coping resource for individuals to manage adverse external stimuli [20]. Low levels of perceived social support may be a risk factor for individuals’ negative emotions and maladjustment [21]. Low levels of perceived social support can increase individual loneliness [22]. Cyberbullying victimization can impair adolescents’ interpersonal relationships and reduce the level of perceived social support. Therefore, cyberbullying victimization may affect loneliness via perceived social support.

### 1.3. Sense of Hope as a Moderator

A sense of hope is an important concept in the field of positive psychology, serving as a buffer for individuals against the adverse effects of negative events (cyberbullying victimization) when they encounter challenging circumstances. Marcel argues that hope serves as an adaptive emotion, particularly when individuals are confronted with negative emotions or adverse circumstances [23]. Stotland argues that hope is an expectation of a goal. The importance and attainability of the goal determine the level of hope [24]. Some studies have shown that a sense of hope can alleviate psychological distress [25] and alleviate loneliness [26]. Research on perceived social support and a sense of hope has revealed a significant dynamic and mutually predictive relationship between the two. Increased levels of perceived social support have been shown to foster the development of a sense of hope. Heightened levels of hope can also enhance an individual’s perceptions of social support [27]. Therefore, the sense of hope may play a moderating role between cyberbullying victimization and perceived social support, as well as between cyberbullying victimization and loneliness.

### 1.4. The Present Study

The current study used a longitudinal research design to examine the effects of T1 cyberbullying victimization on T2 loneliness, as well as the mediating role of T1 social support and the moderating role of T1 sense of hope (see Figure 1). We tested the following hypotheses.

**Hypothesis** **1:**T1 cyberbullying victimization significantly positively predicted T2 loneliness;

**Hypothesis** **2:**T1 cyberbullying victimization may affect middle school students’ T2 loneliness via the mediating effects of T1 social support;

**Hypothesis** **3:**T1 sense of hope moderated the first half and the direct pathway of the mediating effect.

**Figure 1 behavsci-14-00312-f001:**
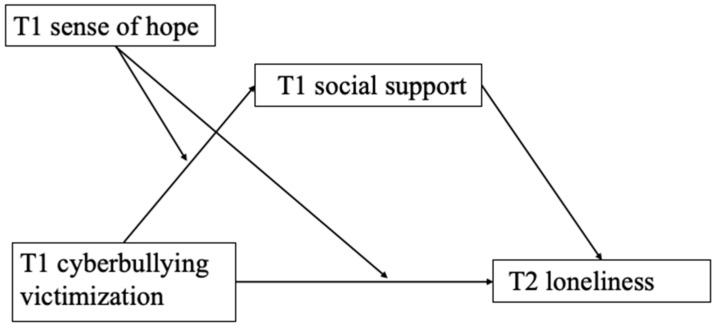
Proposed moderated mediation model.

## 2. Methods

### 2.1. Participants and Procedure

This study was approved by the Ethics Committee of Shaoxing University. The convenience sampling method was used to select students from three middle schools in Shaoxing City for a five-month follow-up survey. The first survey was conducted in May 2023, and the second survey was conducted in October 2023. This survey was completed in the classroom and took about 15 min. Before conducting this survey, participants were briefed on its purpose, the principles of confidentiality, and the intended use of the data. This survey received informed consent from all participants. A total of 656 valid data were collected from the first survey; 599 data were collected from the second survey. Excluding invalid questionnaires (including data that only completed the first survey, data with apparent duplicate responses, and data with a short response time (less than M-2SD)), a total of 583 valid data were collected (females = 51.6%; mean age = 13.34; SD = 0.70).

### 2.2. Measures

#### 2.2.1. Cyberbullying Victimization

The Cyberbullying Victimization Scale developed by Shapka et al. [28,29] and revised by Xie et al. [30] was used. The scale consists of 6 items (e.g., “Received hurtful comments from others about your online photos or videos”.). Each item is scored on a 5-point scale (1 for never, 5 for always). The total score for all items was averaged, with higher scores representing higher levels of cyberbullying victimization. Cronbach’s α = 0.70 for this scale in this study.

#### 2.2.2. Perceived Social Support

The Perceived Social Support Scale developed by Dahlem et al. and revised by Jiang et al. was used [31]. This scale consists of 12 items (e.g., “Some people (teachers, classmates, relatives) are there for me when I encounter something”) and contains three dimensions: family support; friend support; and other support. Each item was scored using a 7-point scale (1 for complete disagreement and 7 for complete agreement). The total scores of all items in this scale were averaged, with higher scores representing higher levels of perceived social support. Cronbach’s α = 0.93 for this scale in this study.

#### 2.2.3. Sense of Hope

The Zhao et al.’s [32] Revised Children’s Sense of Hope Scale was used. This scale consists of 6 items (e.g., “I think I am doing a good job”). It is scored on a 6-point scale (1 for never, 6 for always) and consists of two dimensions: power thinking; and path thinking. The total scores of all questions were averaged, with higher scores indicating higher levels of individual hopefulness. Cronbach’s α = 0.97 for this scale in this study.

#### 2.2.4. Loneliness Scale

The ULS-8 Loneliness Scale [33], revised by Hays and DiMatteo, was used to measure the level of loneliness among middle school students. The scale consists of 8 items (e.g., “I feel left out”). It is scored on a 4-point scale (1 for never, 4 for always), with higher scores indicating higher levels of individual loneliness. Cronbach’s α = 0.73 for this scale in this study.

#### 2.2.5. Covariates

Gender and age can explain individual differences in cyberbullying victimization [5]. Therefore, gender and age were set as control variables in this study.

### 2.3. Data Processing

Data analysis was processed using SPSS 24.0 to examine the relationship between T1 cyberbullying victimization, T1 perceived social support, T2 loneliness, and T1 sense of hope. To avoid the influence of common method bias on the findings, all subjects answered the questionnaire anonymously. First, the Harman one-factor method was adopted to test the common method bias of the questions. The results showed that there were seven factors with eigenvalues >1, the first of which explained 31.80% of the variance, which was less than the critical value of 40%. Therefore, there was no serious common method bias problem in this study. Next, descriptive statistics were analyzed for all variables. Finally, the analysis of moderated mediation effects was performed using SPSS PROCESS model 4 and model 8. The significance of the moderated mediation effect was tested using the bias-corrected bootstrap method.

## 3. Results

### 3.1. Descriptive Statistics

T1 cyberbullying victimization was significantly negatively correlated with T1 perceived social support (*r* = −0.20, *p* < 0.01) and T1 sense of hope (*r* = −0.03, *p* < 0.01) and significantly positively correlated with T2 loneliness (*r* = 0.18, *p* < 0.01). T1 perceived social support was significantly negatively correlated with T2 loneliness (*r* = −0.36, *p* < 0.01) and significantly positively correlated with T1 sense of hope (r = 0.45, *p* < 0.01). T2 loneliness was significantly negatively correlated with T1 sense of hope (*r* = −0.47, *p* < 0.01). The results are shown in Table 1. This result supported Hypothesis 1.

### 3.2. Testing for Mediation Effects

T1 cyberbullying victimization was negatively associated with T1 social support (*β* = 0.35, *p* < 0.001) (Figure 2). SPSS PROCESS Model 4 (Hayes, 2018) was used to test whether T1 social support mediated the link between T1 cyberbullying victimization and T2 loneliness. After controlling for gender and age, results showed (see Table 2) that T1 cyberbullying victimization was negatively associated with T1 social support (*β* = −0.89, *p* < 0.001), which, in turn, was negatively associated with T2 loneliness (*β* = −0.14, *p* < 0.001). The residual direct effect was significant (*β* = 0.21, *p* < 0.01). Thus, T2 social support partially mediated the relationship between T1 cyberbullying victimization and T2 loneliness (indirect effect = 0.06, Boot SE = 0.01, 95% CI = [0.04, 0.10]). The CI does not contain 0, indicating that the mediation effect is significant. This result supported Hypothesis 2 (Figure 3).

### 3.3. Testing for Moderated Mediation Effect

In addition, Model 8 of PROCESS was used to test the moderated mediation hypothesis (Hayes, 2018). As shown in Table 3, in Model 1, there was a significant effect of T1 cyberbullying victimization on T1 social support (*β* = −0.36; *p* < 0.01), and T1 sense of hope moderated this effect (*β* = −0.28; *p* < 0.01). Model 2 showed that the effect of T1 cyberbullying victimization on T2 loneliness was significant (*β* = 0.10; *p* < 0.01), and T1 sense of hope moderated this effect (*β* = −0.18; *p* < 0.001). The effect of T1 social support on T2 loneliness was significant (*β* = −0.07; *p* < 0.001).

A simple slope analysis of the first half of the mediation path is shown in Figure 4. Individuals with higher levels of T1 sense of hope (M + 1SD) also had higher levels of perceived social support compared to those with lower levels of T1 sense of hope (M-1SD) when subjected to the same level of cyberbullying victimization. Bootstrapping analysis of moderating effects found (Table 4) that the effect of T1 cyberbullying victimization on T1 perceived social support was not significant at low levels of sense of hope (CI = [−0.91, 0.10]) and was significant at high levels of sense of hope (CI = [−1.58, −0.72]). The confidence interval does not contain 0, indicating that the moderating effect is significant at high levels of sense of hope. This suggests that T1 cyberbullying victimization on T1 perceived social support is dependent on the moderation of T1 sense of hope.

Simple slope analyses of the direct paths are shown in Figure 5. The relationship between T1 cyberbullying victimization and T2 loneliness became stronger at lower levels of T1 sense of hope (M-1SD) compared to higher levels of T1 sense of hope (M + 1SD). Bootstrapping analysis of moderating effects found (Table 4) that the effect of T1 cyberbullying victimization on T2 loneliness was significant at low levels of sense of hope (CI = [0.33, 0.75]) and not significant at high levels of sense of hope (CI = [−0.13, 0.24]). The confidence interval does not contain 0, indicating that the moderating effect is significant at low levels of sense of hope. This suggests that the effect of T1 cyberbullying victimization on T2 loneliness is dependent on the moderation of T1 sense of hope.

Further Bootstrapping was used to repeat the sampling 2000 times to verify whether the moderated mediation effect was significant. Results showed that the index of moderated mediation effect was 0.019, with 95% confidence interval [0.002, 0.051]. The confidence interval does not contain 0, indicating a significant effect of moderated mediation.

Further Bootstrapping was used to repeat the sampling 5000 times to verify whether the moderated mediation effect was significant. Results showed that the index of moderated mediation effect was 0.019, with 95% confidence interval [0.002, 0.051]. The confidence interval does not contain 0, indicating a significant effect of moderated mediation. This result supported Hypothesis 3.

## 4. Discussion

With the continuous development of society, the network has become an important platform for communication among adolescents. Minors are in the critical stage of physical and psychological development, and bullying in the network will damage their mental health. How to reduce the damage caused by bad behaviors in the network to the physical and mental development of adolescents has become an important topic of concern for researchers. This study employed structural equation modeling within a longitudinal research framework to investigate the impact of cyberbullying victimization on loneliness and its underlying mechanisms among middle school students. It was found that baseline cyberbullying victimization positively predicted the level of loneliness in middle school students, indicating that cyberbullying victimization increased middle school students’ loneliness and harmed their psychological health.

This study found that cyberbullying victimization predicts loneliness and can be mediated by perceived social support. Cyberbullying victimization decreases an individual’s perceived social support, which further decreases an individual’s loneliness. Adolescence is an important stage in psychological development, where seeking social support plays an important role in personality growth and psychological well-being [34]. A study of child maltreatment showed a significant negative correlation between childhood maltreatment and perceived social support [35]. Experiences of emotional and physical maltreatment in childhood can impair adolescents’ ability to perceive social support [36,37]. It has been suggested that perceived social support is an important factor influencing adolescents’ loneliness [11]. A meta-analytic study of adolescent loneliness found that perceived social support was a significant predictor of loneliness [38]. A study of 606 college students found a positive correlation between cyberbullying victimization and depression, with social support mediating the relationship between cyberbullying victimization and depression [39]. The present study further confirmed that perceived social support plays a longitudinal mediating role between cyberbullying victimization and loneliness by developing a longitudinal mediation model. Additionally, cyberbullying victimization can predict individual loneliness levels via the mediating variable of perceived social support.

The present study also found that T1 sense of hope moderated both the direct path and the first half of the mediation process. Specifically, the relationship between cyberbullying victimization and loneliness became stronger among middle school students with low levels of sense of hope compared to those with high levels of sense of hope. In other words, at the same level of cyberbullying victimization, middle school students with high levels of sense of hope experienced less loneliness, whereas middle school students with low levels of sense of hope experienced more loneliness. Recent findings have found hope to be positively associated with youth life satisfaction [40]. A sense of hope mediates the relationship between bullying and students’ emotional difficulties [41]. The sense of hope can alleviate the negative effects of bullying victimization and improve students’ adaptability and subjective well-being [42,43]. A sense of hope helps to reduce individual loneliness [44] and enhance individual life satisfaction and subjective well-being [45]. This study is consistent with the results of the previous studies. A sense of hope plays a positive role in alleviating the negative effects of bullying victimization.

The first half of the pathway of the mediation process was also moderated by the level of a sense of hope. The relationship between cyberbullying victimization and perceived social support became stronger for middle school students with high levels of hope compared to those with low levels of hope. In other words, middle school students with high levels of sense of hope were able to perceive more social support when they suffered the same level of cyberbullying victimization. This shows that a sense of hope can serve as a protective factor for mental health and contribute to enhancing individual self-regulation and self-management abilities. This is consistent with the results of previous studies. Previous studies have shown a positive correlation between social support and hope [46]. Social support can influence individual life satisfaction through hope [47]. Hope is a form of positive motivational force [48] that can help individuals cope with stress and challenges in their lives [49,50]. Research has found that high levels of hopefulness can help individuals be more positive with their families and society [51,52].

This study examined the impact of cyberbullying victimization on loneliness among middle school students and its underlying mechanisms. The findings not only reveal, to some extent, how cyberbullying victimization affects loneliness but also illustrate the change in this effect over time. There is a relative dearth of research on the underlying mechanisms of cyberbullying victimization and its impact on individual mental health in China. This study contributes to filling this gap by enriching the research content in this area, thus making a significant addition to the literature on cyberbullying victimization among adolescents. This study can provide opinions and suggestions for regulating the behavior of social network use and protecting the mental health of adolescents.

This study suggests that parents and schools should pay timely attention to the mental health of students who are victims of cyberbullying and guide them to use social networks correctly. Based on the results of this study, schools can set up student support programs where parents and friends can provide support to cyberbullying victims to alleviate the adverse effects of cyberbullying. Schools can conduct hope interventions to enhance the level of students’ sense of hope and maintain their mental health.

This study still has some limitations that can be improved in future research. First, it is important to note that the protective mechanism of loneliness is complex. While this study primarily examines the roles of perceived social support and sense of hope, other individual protective factors, such as self-esteem and emotion regulation strategies, warrant further investigation. Secondly, it is worth noting that this study only included students from three middle schools. Future research endeavors could aim to broaden the sample range to validate the findings.

## 5. Conclusions

(1)Cyberbullying victimization predicts loneliness;(2)T1 perceived social support mediates between T1 cyberbullying victimization and T2 loneliness;(3)T2 sense of hope moderated the first half and the direct pathway of the mediating effect.

## Figures and Tables

**Figure 2 behavsci-14-00312-f002:**
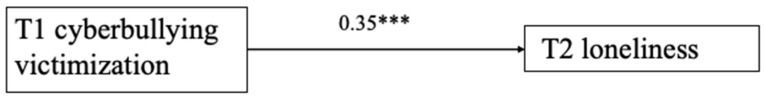
Direct effect. *** *p* < 0.001.

**Figure 3 behavsci-14-00312-f003:**
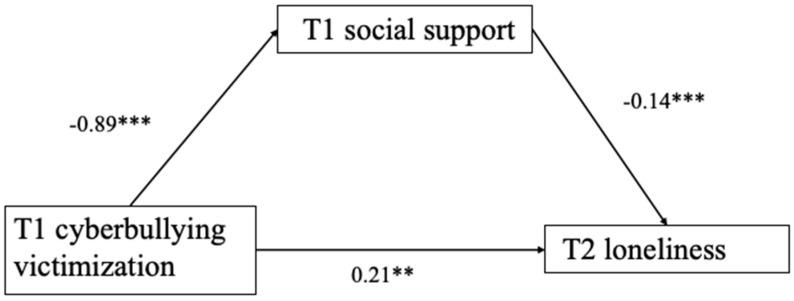
The mediation model. ** *p* < 0.01, *** *p* < 0.001.

**Figure 4 behavsci-14-00312-f004:**
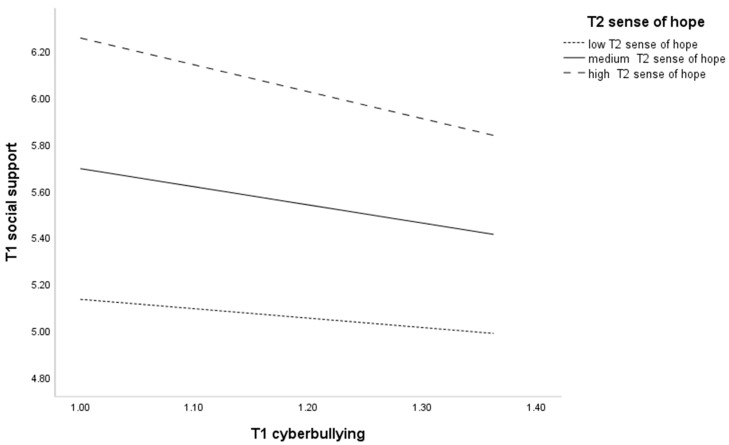
Interaction between T1 cyberbullying and T1 sense of hope on T1 social support.

**Figure 5 behavsci-14-00312-f005:**
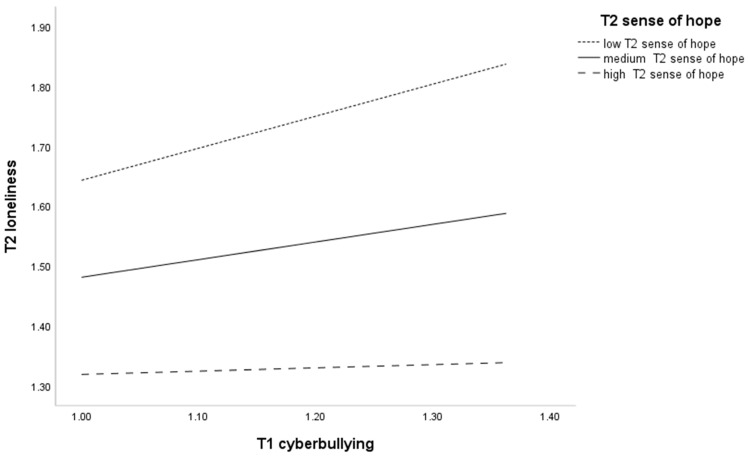
Interaction between T1 cyberbullying and T1 sense of hope on T2 loneliness.

**Table 1 behavsci-14-00312-t001:** Descriptive statistics and correlations of all variables.

Variables	M	SD	1	2	3	4
1. T1 Cyberbullying victimization	1.11	0.25	-			
2. T1 Perceived social support	5.22	1.17	−0.20 **	-		
3. T2 Loneliness	2.09	0.43	0.18 **	−0.36 **	-	
4. T1 sense of hope	2.80	0.57	−0.03	0.45 **	−0.47 **	-

Note: *N* = 583. Gender was dummy variable, encoding 0 = male and 1 = female. ** *p* < 0.01.

**Table 2 behavsci-14-00312-t002:** Testing the mediation effects of T1 cyberbullying victimization on T2 loneliness.

Independent Variables	Model 1 (Criterion T1 Social Support)		Model 2 (Criterion T2 Loneliness)	
	*β*	*t*	*β*	*t*
CO: Gender	−0.14	−1.46	0.05	1.19
CO: Age	−0.12	−1.83	0.10	2.48 *
X: T1 cyberbullying victimization	−0.89	−4.79 ***	0.21	2.76 **
ME: T1 social support			−0.14	−8.37 ***
*R* ^2^	0.05		0.14	
*F*	10.24 ***		23.77 ***	

Note. *N* = 583. All variables were standardized. * *p* < 0.05. ** *p* < 0.01. *** *p* < 0.001.

**Table 3 behavsci-14-00312-t003:** Testing the mediation effects of cyberbullying victimization on loneliness.

Independent Variables	Model 1 (Criterion T1 Social Support)		Model 2 (Criterion T2 Loneliness)	
	*β*	*t*	*β*	*t*
CO: Gender	−0.11	−1.30	0.06	1.76
CO: Age	−0.14	−2.40 *	0.01	0.49
X: T1 cyberbullying victimization	−0.36	−3.39 **	0.10	2.67 **
MEMO: T1 cyberbullying victimization × T1 sense of hope	−0.28	−2.20 **	−0.18	−3.40 ***
ME: T1 social support			−0.07	−3.84 ***
*R^2^*	0.26		0.28	
*F*	39.78 ***		37.24 ***	

Note. *N* = 583. Each column is a regression model that predicts the criterion at the top of the column. Gender was dummy-coded. CO = control variable; X = independent variable; ME = mediator; MEMO = interaction between mediator and moderator. All variables were standardized.* *p* < 0.05. ** *p* < 0.01. *** *p* < 0.001.

**Table 4 behavsci-14-00312-t004:** Direct and mediating effects at different levels of T1 sense of hope.

	T1 Sense of Hope	*β*	SE	95% CI	
				Lower	Upper
Mediating effects of T1 perceived social support	M − 1SD	−0.40	0.26	−0.91	0.10
	M	−0.78	0.17	−1.11	−0.45
	M + 1SD	−1.15	0.22	−1.58	−0.72
Direct effects of T1 cyberbullying victimization	M − 1SD	0.54	0.11	0.33	0.75
	M	0.29	0.07	0.16	0.43
	M + 1SD	0.05	0.09	−0.13	0.24

## Data Availability

The data presented in this study are available on request from the corresponding author.

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
