# Peer review of "The Longitudinal Relationship between Cyberbullying Victimization and Loneliness among Chinese Middle School Students: The Mediating Effect of Perceived Social Support and the Moderating Effect of Sense of Hope"

_behavsci, 2024, doi:10.3390/bs14040312_

Round 1

Reviewer 1 Report

Comments and Suggestions for Authors

Overall comments

Thank you for allowing me to read your paper which I enjoyed. I hope the below comments are useful in strengthening it.

Methods

How was the convenience sample used? What criteria were in place for selecting students from these middle schools? In other words, were all students invited to join the study or did they need to meet certain criteria?

Can you provide some demographic data about the school locations and types of schools e.g. SES, private/public schools etc.   

Can you explain “Excluding invalid 114 questionnaires that were not answered carefully”? why was this decision made?

What ethical issues were apparent in this questionnaire method?

For an international audience can you say how old middle school students are in China and how old the students were in your study?

Discussion

Reference is made to ‘the network’ I assume this is online social network? please can you explain this term?

The discussion is quite short, and more information related to the key results would be helpful. Currently the discussion reiterates the findings with no real engagement with the wider literature. For example, how do these results compare internationally?

The discussion would benefit from considering any policy/practice implications as well as future research directions.

A conclusion summarising the main points from the study is needed.

Author Response

Dear reviewer1:

Thank you for your constructive comments on our manuscript. We have revised our manuscript according to your comments and suggestions. The modifications have been highlighted in red.

Methods

  1. We selected three middle schools in Shaoxing, Chinathat have good cooperative relationships. Because of the heavy learning task of the students in the third grade of middle school, the main participants of this study are the students in the first and second grade. The survey was conducted in class and all participants volunteered to participate.
  2. The three junior high schools are all public schools in Yuecheng District, Shaoxing City, Zhejiang Province, China. The three schools are Shaoxing Wenlan Middle School, Shaoxing Yuanpei Middle School and Shaoxing College Affiliated Middle School.
  3. Excluded data included data that only completed the first survey, data with apparentduplicate responses and data with with a short response time (less than M-2SD).
  4. The apparent ethical issue in this study was the voluntary participation of the participants. Before the beginning of the survey, we told the participants that they could participate in the survey voluntarilyand they had the right to withdraw from the survey at any time during the survey.
  5. Middle school students in China are between the ages of 12 and 15. The middleschool students in this survey were between the ages of 12 and 15, with an average age of 13.34.

Discussion

  1. Online social network is an important platform for teenagers to communicate and exchange information. On social networks, teenagers can post their status, browse other people's life information and communicate and interact with others.WeChat, Weibo, and QQ are the most popular social networks used by Chinese teenagers.
  2. The discussion section adds a comparison between the results of this study and the existing literature. Expanded the discussion section.
  3. The discussion increases policy/practice implications of this study.
  4. The conclusion is added, and the main points of this study are expounded.

Once again, we would like to express our gratitude to you for your invaluable feedback and constructive comments, which have contributed significantly to the improvement of our manuscript. We hope that the revised version of our manuscript meets with your approval.

Sincerely yours,

Jing Wu

Reviewer 2 Report

Comments and Suggestions for Authors

Review Summary:

(1) This is a sample in which low levels of cyberbullying victimization can be seen N~(1.11; .25), min: 1; max: 5. In the description of the results (descriptive statistics) it may be of interest to know the subjects with the highest levels of cyberbullying victimization, the subjects with the lowest levels of cyberbullying victimization.

(2) The longitudinal nature of the analysis should be clarified.

(3) The presentation of the results was difficult to understand. It would be interesting to make the presentation more accessible to readers.

(4) It would be worthwhile to expressly include the conditions of mediation and moderate mediation, illustrating the results obtained.

Author Response

Dear reviewer 2:

Thank you for your constructive comments on my manuscript. We have revised our manuscript according to your comments and suggestions. The modifications have been highlighted in red.

  1. Descriptive statistics include descriptions of the mean of variables, standard deviations, and correlations between variables. In the discussion section, the different moderating effects of high level cyberbullying victimization and low level cyberbullying victimization were analyzed.
  2. This study used mediated moderation models, which were analyzed using SPSS PROCESS (Hayes, 2018). Firstly, SPSS PROCESS Model 4 was used to test the mediation hypothesis. Then, SPSS PROCESS Model 8 was used to test the moderated mediation hypothesis. Mediated moderation models are often used to analyze relationships between variables in cross-sectional studies, and this study used longitudinal data to illustrate the relationship between variables over time. This research model has been used in previous studies (Precht et al., 2022; Xiang et al., 2022; Wu et al., 2021; Sarwar et al., 2021)
  3. We have modified the presentation of the results to make the article easier to read. The changes have been highlighted in red.

The effects of gender and age were not significant in this study.

  1. The changes have been highlighted in red.

Reference

Precht, L. M., Stirnberg, J., Margraf, J., & Brailovskaia, J. (2022). Can physical activity foster mental health by preventing addictive social media use?–A longitudinal investigation during the COVID-19 pandemic in Germany. Journal of affective disorders reports, 8, 100316.

Xiang, G. X., Gan, X., Jin, X., Zhang, Y. H., & Zhu, C. S. (2022). Developmental assets, self-control and internet gaming disorder in adolescence: testing a moderated mediation model in a longitudinal study. Frontiers in Public Health, 10, 808264.

Wu, N., Hou, Y., Zeng, Q., Cai, H., & You, J. (2021). Bullying experiences and nonsuicidal self-injury among Chinese adolescents: a longitudinal moderated mediation model. Journal of youth and adolescence, 50(4), 753-766.

Sarwar, A., Bashir, S., & Karim Khan, A. (2021). Spillover of workplace bullying into family incivility: testing a mediated moderation model in a time-lagged study. Journal of interpersonal violence, 36(17-18), 8092-8117.

Once again, we would like to express our gratitude to you for your invaluable feedback and constructive comments, which have contributed significantly to the improvement of our manuscript. We hope that the revised version of our manuscript meets with your approval.

Sincerely yours,

Jing Wu

Reviewer 3 Report

Comments and Suggestions for Authors

Comments on the Quality of English Language

Moderate editing.

Author Response

Dear reviewer 3:

Thank you for your constructive comments on my manuscript. We have revised our manuscript according to your comments and suggestions. The modifications have been highlighted in red.

  1. Chinese middle school students have a five-month semester. We collected data for the first time at the beginning of the semester and for the second time at the end of the semester. We also referred to the relevant literature (Camerini et al., 2020; Wang et al., 2022; Awadalla et al., 2020) that it is feasible to collect data twice at an interval of five months.

Reference

Camerini, A. L., Marciano, L., Carrara, A., & Schulz, P. J. (2020). Cyberbullying perpetration and victimization among children and adolescents: A systematic review of longitudinal studies. Telematics and informatics, 49, 101362.

Wang, W., Guo, Y., Du, X., Li, W., Wu, R., Guo, L., & Lu, C. (2022). Associations between poor sleep quality, anxiety symptoms, and depressive symptoms among Chinese adolescents before and during COVID-19: a longitudinal study. Frontiers in Psychiatry, 12, 786640.

Awadalla, S., Davies, E. B., & Glazebrook, C. (2020). A longitudinal cohort study to explore the relationship between depression, anxiety and academic performance among Emirati university students. BMC psychiatry, 20, 1-10.

  1. This study used mediated moderation models, which were analyzed using SPSS PROCESS (Hayes, 2018). Firstly, SPSS PROCESS Model 4 was used to test the mediation hypothesis. Then, SPSS PROCESS Model 8 was used to test the moderated mediation hypothesis. Mediated moderation models are often used to analyze relationships between variables in cross-sectional studies, and this study used longitudinal data to illustrate the relationship between variables over time. This research model has been used in previous studies (Precht et al., 2022; Xiang et al., 2022; Wu et al., 2021; Sarwar et al., 2021)

Reference

Precht, L. M., Stirnberg, J., Margraf, J., & Brailovskaia, J. (2022). Can physical activity foster mental health by preventing addictive social media use?–A longitudinal investigation during the COVID-19 pandemic in Germany. Journal of affective disorders reports, 8, 100316.

Xiang, G. X., Gan, X., Jin, X., Zhang, Y. H., & Zhu, C. S. (2022). Developmental assets, self-control and internet gaming disorder in adolescence: testing a moderated mediation model in a longitudinal study. Frontiers in Public Health, 10, 808264.

Wu, N., Hou, Y., Zeng, Q., Cai, H., & You, J. (2021). Bullying experiences and nonsuicidal self-injury among Chinese adolescents: a longitudinal moderated mediation model. Journal of youth and adolescence, 50(4), 753-766.

Sarwar, A., Bashir, S., & Karim Khan, A. (2021). Spillover of workplace bullying into family incivility: testing a mediated moderation model in a time-lagged study. Journal of interpersonal violence, 36(17-18), 8092-8117.

  1. The survey was conducted in class, students had enough time to give answers, and students gave serious answers, so the Cronbach's α was relatively high.

Once again, we would like to express our gratitude to you for your invaluable feedback and constructive comments, which have contributed significantly to the improvement of our manuscript. We hope that the revised version of our manuscript meets with your approval.

Sincerely yours,

Jing Wu

Round 2

Reviewer 2 Report

Comments and Suggestions for Authors

The authors have made the proposed modifications.

In a similar way to the two figures that illustrate (1) the relationship between cyberbullying victimization and loneliness, and (2) the mediation of social support in the relationship between the two previous variables, it could have improved the reading of the work to have introduced a figure that illustrate the moderated mediation relationship.

See review report v1 figure: Model moderated mediation. Statistical Diagram, or figure at the end of the MODERATE MEDIATION HYPOTHESIS section (page 4 of 5).